# Use of Spent Coffee Ground as an Alternative Fuel and Possible Soil Amendment

**DOI:** 10.3390/ma15196722

**Published:** 2022-09-27

**Authors:** Lukáš Jeníček, Barbora Tunklová, Jan Malaťák, Michal Neškudla, Jan Velebil

**Affiliations:** 1Faculty of Engineering, Czech University of Life Sciences Prague, Kamýcká 129, 165 00 Prague, Czech Republic; 2Faculty of Agrobiology, Food and Natural Resources, Czech University of Life Sciences Prague, Kamýcká 129, 165 00 Prague, Czech Republic

**Keywords:** biomass, biofuel, calorific value, spent coffee ground, phytotoxicity

## Abstract

Spent coffee ground is a massively produced coffee industry waste product whose reusage is beneficial. Proximate and ultimate and stochiometric analysis of torrefied spent coffee ground were performed and results were analyzed and compared with other research and materials. Spent coffee ground is a material with high content of carbon (above 50%) and therefore high calorific value (above 20 MJ·kg^−1^). Torrefaction improves the properties of the material, raising its calorific value up to 32 MJ·kg^−1^. Next, the phytotoxicity of the aqueous extract was tested using the cress test. The non-torrefied sample and the sample treated at 250 °C were the most toxic. The sample treated at 250 °C adversely affected the germination of the cress seeds due to residual caffeine, tannins and sulfur release. The sample treated at 350 °C performed best of all the tested samples. The sample treated at 350 °C can be applied to the soil as the germination index was higher than 50% and can be used as an alternative fuel with net calorific value comparable to fossil fuels.

## 1. Introduction

Coffee has a long history and wide usage being consumed for over 1000 years. In 2020, the total global production of coffee was 175 million 60 kg bags, which is equal to 10.5 megatons of coffee [1]. The consumption was very similar: 166 million 60 kg bags, or 9.9 megatons [2,3]. If we consider 9 g of coffee used for the preparation of one coffee cup, the coffee consumption increased from 400 billion cups in 2011 [4] to more than 1 trillion in 2021. By no accident is coffee the most consumed drink and second most traded commodity after petroleum today [5]. Numerically, 1 ton of green coffee generates about 650 kg of spent coffee ground (SCG) [6,7] which gives a worldwide annual generation of 6.5 megaton of SCG.

There were approximately 3 billion coffee cups drunk in Czech Republic in 2019, which produced around 24,000 tons of SCG [8]. Households can directly combust this material [9], use it as a fertilizer for their gardens or leave it to be collected by waste disposal companies, which may let the material be improved by torrefaction and use as part of industrial processes.

Coffee is very rich in minerals. Coffee beans are composed of many components, including cellulose, sugars, minerals, lipids, polyphenols and tannin [10,11,12,13]. Minerals include magnesium, potassium, calcium, iron, sodium, nickel, manganese, strontium, barium, chromium, rubidium, zinc, copper, vanadium, cobalt, molybdenum, titanium, lead and cadmium [14,15,16].

SCG can also be used as a potential source to produce biodiesel or fuel pellets [17,18,19,20,21,22]. Colantoni described SCG as an excellent raw material, with great values in terms of low ash content and high calorific value, allowing the production of almost pure (98%) SCG pellets that are suitable for thermal conversion system use [23]. SCG can also be used as fuel in industrial boilers due to its high calorific value of approximately 20.9 MJ·kg^−1^, which is comparable with other agro-industrial materials [24,25,26,27,28]. SCG, as a carbon-rich by-product, may be converted into value-added biopolymer [28].

Torrefaction is a thermochemical process of biomass being slowly heated to be converted into coal-like material [29]. The results of proximate and proximate and ultimate analysis showed positive influence of torrefaction on the samples [30,31]. SCG can be easily converted by torrefaction to a high-value fuel product due to the high concentration of sugars, fibers, proteins and compounds such as fatty acids, aldehydes, ketones, alcohols, hemicellulose, cellulose or lignin [32,33].

Biochar production from biomass feedstocks is a reasonable approach to climate change mitigation of greenhouse gas and waste management [34,35,36]. Potential utilization of the generated residues for fuels or high value-added chemicals production by means of pyrolysis results in, at least, an attractive and challenging solution for SCG, whose worldwide production keeps increasing [37,38]. On the other hand, it was observed that with the increase in pyrolysis temperature and heating rate, the biochar yield declined. The maximum conversion yield in the biochar was measured at 400 °C [39,40].

Torrefaction changes the physical properties of the material by compromising the fiber strength and facilitating comminution and thus improves and accelerates the co-firing of biomass in coal power plants [41,42]. Therefore, the torrefaction of biomass may enable use of sustainable fuels without the need for additional installations in coal power plants leading to lower coal consumption [43].

The conventional energy sources, such as fossil fuels, are today serving almost 81.7% of world energy needs [44]. The energetic potential of spent coffee grounds is lower than that of fossil fuels, but represents a higher energy content in comparison to other biomasses [45,46].

The physico-chemical properties of biochar, used as a renewable energy source, are essential for the optimalization of its production use [47,48,49]. Moisture and ash content are important parameters of the biochar quality, which precise determination is essential [50] as well as accurate determination of calorific value [51]. The fraction of individual elements (CHNS) has a major influence on the thermal output of the combustion plant [52], but also on the resulting emission concentrations during combustion of biochar fuels [9].

At a consumer level, SCG can be used as garden fertilizer as it is high in nutrients [45]. However, as an agro-waste material which is typically discarded in landfills or incinerated, it causes severe environmental problems due to the disintegration and potential release of toxic and residual caffeine tannin, lipids contaminants or methane under anaerobic conditions and, thus, contributes to adverse impact on the environment and climatic change [39].

Most studies have found that biochar application improves soil fertility, increases crop yields, reduces greenhouse gas emissions and increases soil carbon stocks [53,54]. It was further demonstrated that biochar can be used as a soil additive because its structure allows it to bind and retain water in the soil. It also allows better aeration and promotes microbial activity and nutrient transfer [55,56]. Due to its properties and stable organic form, biochar can be an attractive material in the field of agriculture. Biochar increases the water retention capacity of the soil or substrate. It lightens the soil, raises the pH, increases microbial activity and at the point of application, increases the use of elements derived from fertilizers. By retaining and gradually releasing them, it also reduces their leakage into groundwater [56,57]. Biochar is highly stable to decomposition, so it can remain in the soil for a longer time and gradually release substances into the soil. Another advantage is its ability to retain nutrients and bind water. However, the physical and chemical properties of biochar depend on pyrolysis conditions such as temperature and feedstock type [57]. The recommended application rate for any soil treatment should be based on extensive laboratory, greenhouse and field experiments. Currently, there is insufficient data to make general recommendations. In addition, biochar properties may vary depending on the feedstock (e.g., pH, ash content, moisture and nutrient content). This also affects the application rate. Several studies have reported positive effects of biochar application on crop yields ranging from 5 to 50 tons per hectare with appropriate nutrient management [57,58,59].

Biochar for soil application must meet certain criteria to be safe. According to the European Biochar Certificate, the H/C ratio, which describes the stability of the substance, must not exceed 0.7, and the O/C ratio must be less than 0.4. In addition, the presence of nutritionally important elements (N, P, Mg, K, Ca) and heavy metals (Pb, Cd, Cu, Ni, Hg, Zn, Cr, As) is monitored. The content of polycyclic aromatic hydrocarbons PAHs shall not exceed the limits set [60].

There has been much research conducted in the field of SCG and torrefaction, but so far, no research has combined an in-depth analysis of SCG torrefied on several heat levels for usage as a fuel and soil amendment. The goal of this paper is to evaluate the applicability of SCG and its biochar as a direct combustion alternative or as a soil amendment reaching the combustion and germination properties of other materials.

The hypothesis for this paper is that SCG can be used as an alternative for fossil fuels and an alternative soil amendment, and the torrefaction of the material increase the energy yield and enhances the material properties.

## 2. Materials and Methods

Spent Coffee Ground (SCG) was chosen as a material for this research. Specifically, mixture of *Coffea arabica*, one of 120 individual coffee species, covering 75% of world’s production was chosen. The botanical genus and species name for *Coffea arabica*, also written as *C. arabica* originated in the forests of Ethiopia and South Sudan. *C. arabica* was then spread throughout the world to produce its seeds [61].

A total amount of 3 kg of SCG was collected by one household over a period of 4 months. The SCG was dried naturally on the sun reaching the moisture mass share at the level of 8.22%.

For the purpose of the research, material was modified by torrefaction in a programmable weighing furnace LECO TGA 701 at a rate of 10 °C·min^−1^ up to the defined temperature which was then maintained for another 60 min. See the list of analyzed samples in Table 1.

### 2.1. Proximate and Ultimate Analysis

Proximate and ultimate analysis was performed for each sample by measuring the water and ash content, combustion heat and elemental composition.

The moisture content was determined by Thermogravimetric analyzer LECO TGA701 where 1 g of each sample was dried at a temperature of 105 °C until the weight became constant.

The ash content was also determined by Thermogravimetric analyzer LECO TGA701 where 1 g of each sample was heated with increased oxygen concentrations up to temperature of 550 °C until the weight became constant.

Combustion heat was measured by an isoperibolic calorimeter LECO AC600 by controlled combustion in a bomb under high pressure of 3 MPa at a reference temperature of 28 °C. Stainless steel cups were used for the material placement and cotton thread for ignition. The device was calibrated with benzoic acid.

The element composition was measured with an elemental analyzer LECO CHN628 + S using the LECO instrumental combustion method for biomass. To determine the C, H and N values, a 0.1 g of each sample was burned in oxygen at the temperature of 950 °C. The analyzer was calibrated with ethylenediaminetetraacetic acid, rice and rye flour. Oxygen was determined as a difference in the dry sample.

### 2.2. Stochiometric Analysis

Stoichiometric calculations are an important basis for thermal analysis. They are important for controlling the operation of combustion plants and are necessary for comparison of the actual and theoretical combustion processes. They make use of the results of previous analyses [62]. The reference amount of oxygen for calculations was set as 10% following the Czech legislation for biomass combustion.

#### These Calculations Determine

Calorific value influenced by material moisture:

The calorific value was calculated from the results of ultimate and proximal analyses of individual samples.

The relationship between gross calorific value $Q_{s}=kJ\cdot kg^{-1}$ and net calorific value $Q_{i}=kJ\cdot kg^{-1}$ [63].

The theoretical amount of oxygen consumption for complete combustion *O*_2,_*_min_* (m^3^ kg^−1^) was based on the equation:(1)$$Om\left(O_{2}\right){\left(\frac{C}{M\left(C\right)}+\frac{H}{M\left(2\cdot H_{2}\right)}+\frac{S}{M\left(S\right)}-\frac{O}{M\left(O_{2}\right)}\right)}_{2,min}\;$$
where *C*, *H*, *S* and *O* are contents of carbon, hydrogen, sulfur and oxygen, respectively, in the sample (% wt.), *V_m_*(*O*_2_) = 22.39 m^3^ kmol^−1^ is the molar volume of oxygen gas at normal conditions and *M*(*X*) (kg kmol^−1^) are molar masses of hypothetical species *X* that combine with *O*_2_.

The theoretical amount of dry combustion air *L_min_* (m^3^_N_ kg^−1^) was determined from the equation:(2)$$L\frac{100}{C_{atm}\left(O_{2}\right)}2,min_{min}$$
where *C_atm_*(*O*_2_) = 23.20% vol. is mass concentration of oxygen in air.

The theoretical amount of dry flue gas *v_fg,min_* (kg·kg^−1^) was calculated by the equation:(3)$$v\frac{V_{m}\left(CO_{2}\right)}{M\left(C\right)}\frac{V_{m}\left(SO_{2}\right)}{M\left(S\right)}\frac{V_{m}\left(N_{2}\right)}{M\left(N_{2}\right)}{\frac{C_{atm}\left(N_{2}\right)}{100}}_{{min}_{fg,min}}$$
where *V_m_*(*X*) (kg.kmol^−1^) was the molar mass of flue gas components, *C_atm_*(*N*_2_) = 75.474% vol. is the concentration of *N*_2_ in air.

The theoretical amount of emission concentrations of *CO*_2,_*_max_* (kg·kg^−1^) was based on the equation:(4)$$CO{\frac{M\left(C\right)\cdot C}{V_{m}\left(CO_{2}\right)\cdot v_{fg,min}}}_{2max}$$

Volumetric amounts of combustion products:(5)$$v\left(CO_{2}\right)=\frac{V_{m}\left(CO_{2}\right)}{M\left(C\right)}\cdot C+\frac{C_{atm}\left(CO_{2}\right)}{100}\cdot L$$
(6)$$v\left(SO_{2}\right)=\frac{2\cdot V_{m}\left(SO_{2}\right)}{2\cdot M\left(S\right)}\cdot$$
(7)$$v_{N_{2}}=\frac{V_{m}\left(N_{2}\right)}{M\left(N_{2}\right)}\cdot N+O{\frac{C_{atm}\left(N_{2}\right)}{C_{atm}\left(O_{2}\right)}}_{2,min}$$

Conversion of the calorific value of *Q_i_* at an arbitrary water content *W* to a different water content *Wt* is made according to the formula:(8)$$Q_{in}^{}=\frac{100-Wt}{100-W}\cdot \left(Q_{i}^{}+0.02442\cdot W\right)-0.02442\cdot Wt$$
where *Wt* (% wt.) is the total water content in the original sample; *W* is the net calorific value of the original sample (MJ·kg^−1^); and *Q_i_* is the net calorific value at the target water content.

### 2.3. Phytotoxicity Test

A phytotoxicity test on model plant of garden cress (*Lepidium sativum* L.) was performed by the department of Botany and Plant Physiology, Czech University of Life Sciences Prague according to [64] with modification.

#### 2.3.1. Preparation of the Extract

Ten grams of studied material was placed in a suitable container to which distilled water was added. The amount of distilled water was determined by the amount of dry matter in the sample, the volume of water being 10 times the dry matter content. The container with the biochar/water mixture was placed on a horizontal shaker. Two hours later, the sample material was infused into the distilled water. Finally, the material was filtered through a 15-μm-pore size (Papírny Perštejn KA-0—qualitative filter paper) via a 12-cm-diameter Buchner funnel. The filtrate was then used for the cress test. The resulting solution concentration was based on the methodology and a 10% solution was always used.

#### 2.3.2. Seed Germination Test

*Lepidium sativum* L. seeds are used for germination and phytotoxicity tests due to their specific properties such as rapid growth, germination and high sensitivity to toxic substances [65].

A filtration paper was placed in 11 cm diameter Petri dishes to cover the bottom of the dish. The paper was moistened with 5 mL of pipetted aquaeous extract, or distilled water for a control sample. Thirty garden cress seeds were evenly placed on the filtration paper. Five Petri dishes were prepared for each testing variant, followed by five Petri dishes for the control. The Petri dishes were sealed with parafilm and placed in an incubator. The seeds were germinated in the incubator for 48 h in complete darkness at 25 °C. A seed was considered germinated when the radicle was longer than 2 mm.

The germination index, which is an indicator of biochar toxicity, was calculated using:$$GI=\frac{k_{v}.l_{v}}{k_{k}.l_{k}}.100\;\;\left(\%\right)$$

*k_v_*–germination of the sample

*k_k_*–germination of the control variant

*l_v_*–average root length of the sample (mm)

*l_k_*–average root length of the control (mm)

The germination index of at least 50% is required for use in the soil [66,67].

#### 2.3.3. Statistical Analysis

Analysis of variance (ANOVA) and Tukey’s test were used to evaluate the results using STATISTICA 12.0 CZ software Statsoft, Tulsa, OK, USA) at 0.05 level of significance.

## 3. Results and Discussion

### 3.1. Proximate and Ultimate Analysis

The average results of proximate and ultimate analysis are shown in Table 2 and visualized in Figure 1. The main components of the original sample SCG0 are carbon (50.26% wt.) and oxygen (31.37% wt.). The moisture had the share of 8.22% wt. and ash share was at a level of 1.59% wt. The net calorific value of SCG0 was 19.74 MJ·kg^−1^, which is in line with the findings of Silva [26], who set the calorific power of approximately 5000 kcal·kg^−1^, which is 20.92 MJ·kg^−1^, as well as Colantoni [23], who set the calorific value at 22.36 MJ·kg^−1^.

When torrefied at 250 °C, the original moisture level of 8.22% wt. in SCG0 decreased to only 0.86% wt. The moisture bounced slowly back with a torrefaction temperature increase up to 6.17% wt. for SCG550. Fermoso [37] measured the moisture at the amount of 5% wt. for SCG0.

Together with moisture, the hydrogen and oxygen level also decreased with the increase in torrefaction temperature, as expected [24,68]. The hydrogen level decreased from 6.29% wt. for SCG0 to 2.60% wt. for SCG550 and the oxygen level decreased from 31.37% wt. for SCG0 to 3.17% wt. for SCG550.

On the contrary, carbon, nitrogen and ash levels increased with torrefaction temperature rise. Carbon, the main power source of the material, increased its share from 50.26% wt. for SCG0 up to 77.94% wt. for SCG450. Fermoso and Mašek [37] measured the carbon share at a very similar level of 53.90% wt., as well as Mayson [45], who measured the carbon share in SCG at the level of 53.32% wt.

The ash share increased from 1.59% wt. for SCG0 to 6.95% wt. for SCG550. The nitrogen share increased from 2.21% wt. for SCG0 to 4.41% wt. for SCG450. Ash content results were slightly higher than measured by Colantoni [23] whose values did not exceed 1.30% wt. Common solid wood biomass ash share is usually 0.1–8.4% wt. depending on the quality of woody biomass [69].

The amount of carbon in spent coffee ground influences directly the level of combustion heat and calorific value of the material. As can be seen in Figure 2, the carbon share increase led to an increase in net calorific value in material up to sample SCG350. By SCG450, a drop in net calorific value can be seen, even though the carbon share was still on higher level. The calorific value increased from 19.74 MJ·kg^−1^ for SCG0 up to 31.26 MJ·kg^−1^ for SCG350. This drop in net calorific value may be explained by the decrease in the share of hydrogen, which is an important factor in combustion processes.

From the list of other biomasses, orange peel, analyzed by Tamelova [30], contains a calorific value of 24.97 MJ·kg^−1^ at the biochar of 275 °C, which is approximately 10% higher than for similar biochar of SCG. Fermented Palm Oil reached the highest calorific value of 21.25 MJ·kg^−1^ [70]. Wood with an average moisture of 30–45% contains, in comparison, a very low calorific value of 9–12 MJ·kg^−1^ [29]. Coal contains approximately 23–28 MJ·kg^−1^, which is similar value as for SCG300 and SCG350.

### 3.2. Stochiometric Analysis

Various stochiometric analyses were conducted to better understand the sample behavior under different conditions, which are essential for the optimal setup of thermochemical processes [71].

The net calorific value is indirectly influenced by material moisture. Higher moisture levels lead to a lower net calorific value of a material [72]. Figure 3 shows net calorific value and the moisture levels of different materials. Out of all analyzed materials, SCG350 showed the highest net calorific value for all levels of moisture, up to 32 MJ·kg^−1^ with zero moisture in the material. The lowest net calorific value is present in the original SCG0 sample with 21 MJ·kg^−1^ with zero moisture.

The amount of mass flow of fuel led into combustion chamber is directly influenced by the desired combustion plant heat output (kW). Higher heat output needs more fuel to be combusted. SCG350 with the highest calorific value needs the least amount of mass flow of all analyzed materials. To gain 260 kW of a power, 33.30 kg of SCG350 needs to lead into combustion chamber every hour. On the contrary, to gain the same output of 260 kW, 53.20 kg of SCG0 needs to lead to combustion chamber if we use the original material. Detail of the mass flow led into combustion chamber with the heat output (kW) is shown in Figure 4.

The dependence between O_2_ and CO_2_ is:y=−0.901x+17.618

A detailed result for all analyzed samples is shown on Figure 5. Highest CO_2_ concentrations were calculated for SCG450 and SCG550. The figure shows the combustion process as a function of the oxygen concentration in the flue gas. A zero-oxygen concentration in the flue gas indicates perfect combustion. Each point on each curve represents a given excess air coefficient, which takes the value of 1 at zero oxygen concentration in the flue gas and gradually increases to 5.5 for higher oxygen concentrations. A higher level of CO_2_ signifies higher combustion efficiency of combustion plant and is, therefore, desired.

The theoretical combustion air amounts for perfect combustion are shown on Figure 6.

The theoretical amount of air for perfect combustion *L_min_* (kg·kg^−1^) increased together with the torrefaction degree up to SCG350. There is a slight decrease for SCG450 and SCG550. The same trend has the theoretical mass amount of dry flue gas *v_fg,min_* (kg·kg^−1^) with the highest amount for SCG350. As the amount of air for perfect combustion increases, the CO_2_ mass concentration in dry flu gas decreases, CO_2*max*_ (% wt). The decrease in oxygen with the temperature increase caused the CO_2*max*_ (% wt.) decrease for SCG300 and SCG350. Meanwhile, the increase in carbon then caused the turnback of CO_2*max*_ (% wt.) for SCG450 and SCG550.

Very similar numbers are seen also on the Figure 7 showing the theoretical volumetric combustion. These results are consistent with Tamelová [24] and Jeníček [73], where a higher torrefaction temperature reduced the fraction of CO_2_ concentration in the flue gas.

### 3.3. Phytotoxicity Test

Torrefied SCG proved to be an applicable soil amendment in a phytotoxicity test. There is a germination index comparison between the control sample and analyzed samples in Figure 8. Control sample is set to the level of 100%. The original SCG0 sample germinated at the level of 20% of the control sample only. SCG0 as a crude plant waste is characterized by a strong phytotoxic effect, associated, among others, with its content of caffeine, tannins and polyphenols [74]. According to Cervera-Mata [75], phenolic compounds may be responsible for the growth inhibition of SCG. Regarding the nature of potentially toxic polyphenols in SCG, Jiménez-Zamora [76], reported that the content of total polyphenols in spent coffee grounds is around 17.30% wt. (e.g., chlorogenic acid). According to Griffith [77] other organic compounds present in SCG may inhibit plant germination or growth. SCG0 is, therefore, not a beneficial soil amendment.

For the SCG350 sample, the content of germination index increased by 11% compared to the control. This test shows that the SCG300, SCG350, SCG450 and SCG550 samples tested can be applied to the soil after torrefaction. This is in accordance with the work of Hejátková [64], which states that for the possible use of the investigated sample to soil, a germination index of at least 50% is required. Furthermore, our research shows that biochar produced at a higher temperature of 350–550 °C had fewer phytotoxic effects than biochar produced at the temperature of 250 °C. These results correspond with Ronsse [78], where higher torrefaction temperature reduced the phytotoxicity of biochar.

Our results show that the torrefied sample SCG250 resulted in an increase in phytotoxicity. It is due to the formation of undesirable toxic substances by heat treatment of the sample as residues of caffeine, tannins and sulfur.

## 4. Conclusions

Results of proximate and ultimate and stochiometric analysis confirmed the properties of spent coffee ground found by other researchers. Ash content, which was approximately 20% higher then found by other authors, was the only difference measured. The properties of spent coffee ground showed the ability of the material to be used as a biofuel reaching the average net calorific value of 20 MJ·kg^−1^, which is comparable amount to other biofuels. Biochar from spent coffee ground prepared by torrefaction for 60 min at the temperature of 350 °C improved the net calorific value up to 32 MJ·kg^−1^. At this level, net calorific value is even comparable to fossil fuels where coal gains approximately 23–28 MJ·kg^−1^. Additional experiments need to be performed to evaluate if spent coffee ground can be used as an alternative to coal in coal power plants without any necessary plant adjustments or as a coal additive. The results of the phytotoxicity test show that the non-torrefied SCG samples and the samples torrefied at 250 °C are toxic and unsuitable for use in soil amendment. The heat treatment at higher temperatures breaks down various naturally occurring substances, which can inhibit seed germination. The sample treated at 350 °C can be applied as a soil amendment. It had the best parameters of germination in phytotoxicity test. As a result, spent coffee ground torrefied for 60 min at the temperature of 350 °C is a valuable option to use as a biofuel or soil amendment; therefore, the initial hypothesis can be confirmed.

## Figures and Tables

**Figure 1 materials-15-06722-f001:**
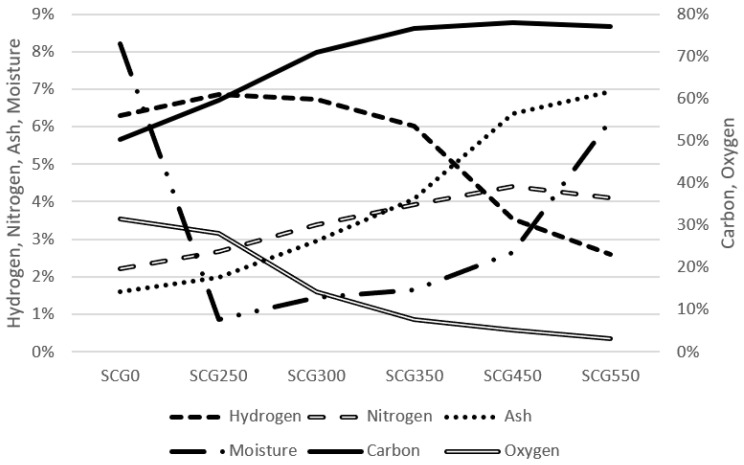
Coffee proximate and ultimate analysis.

**Figure 2 materials-15-06722-f002:**
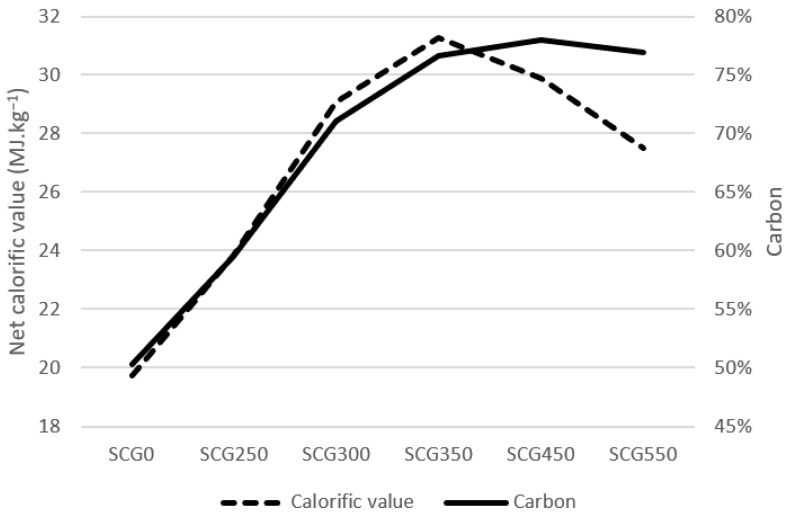
Carbon and net calorific value mutual influence.

**Figure 3 materials-15-06722-f003:**
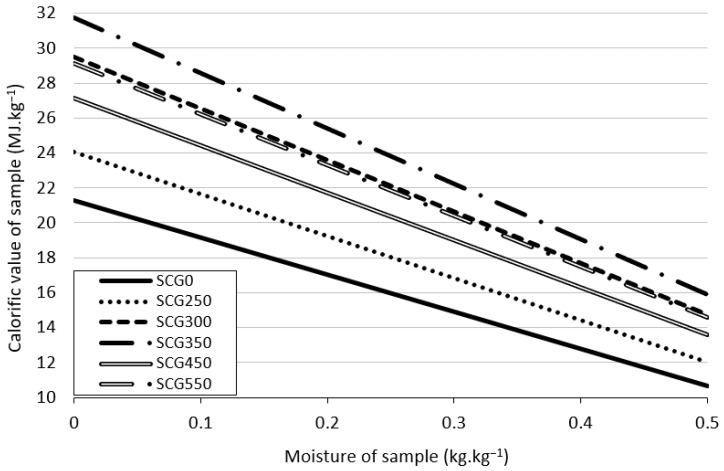
Net calorific value influenced by material moisture.

**Figure 4 materials-15-06722-f004:**
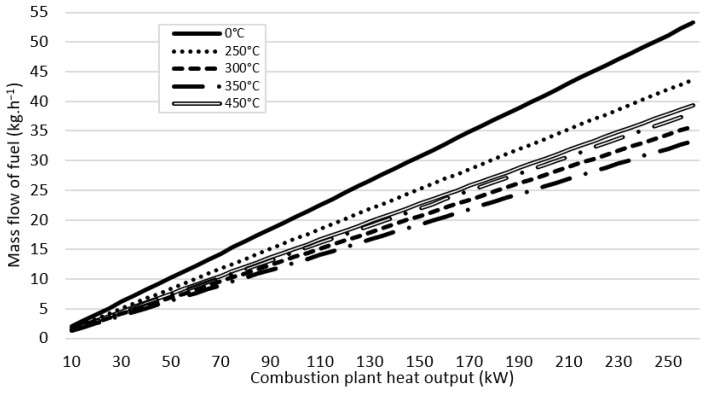
Mass flow of fuel fed into the combustion chamber to reach the combustion heat output (kW).

**Figure 5 materials-15-06722-f005:**
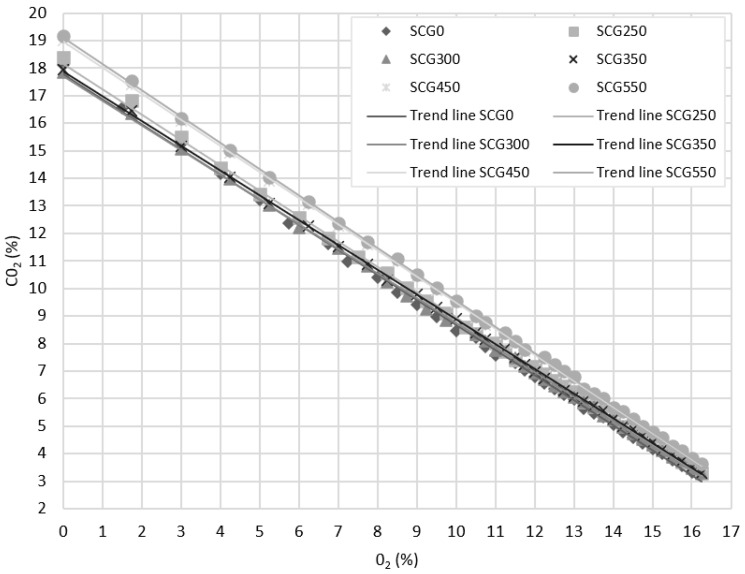
Coffee oxygen share dependence on carbon dioxide share.

**Figure 6 materials-15-06722-f006:**
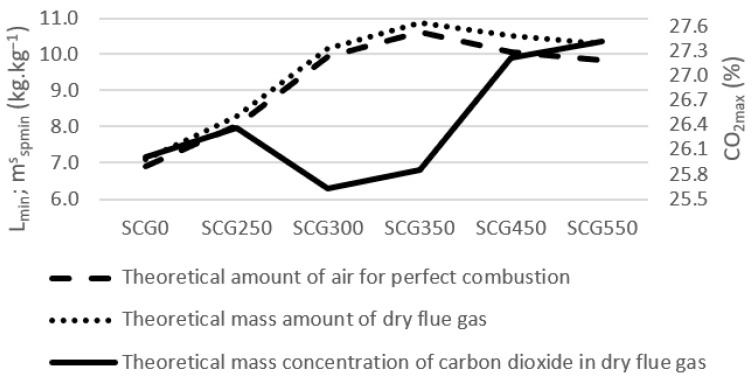
Mass combustion of spent coffee grounds.

**Figure 7 materials-15-06722-f007:**
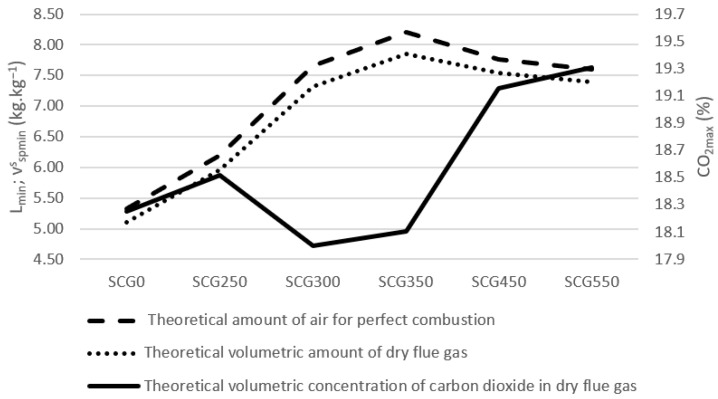
Volumetric combustion of spent coffee grounds.

**Figure 8 materials-15-06722-f008:**
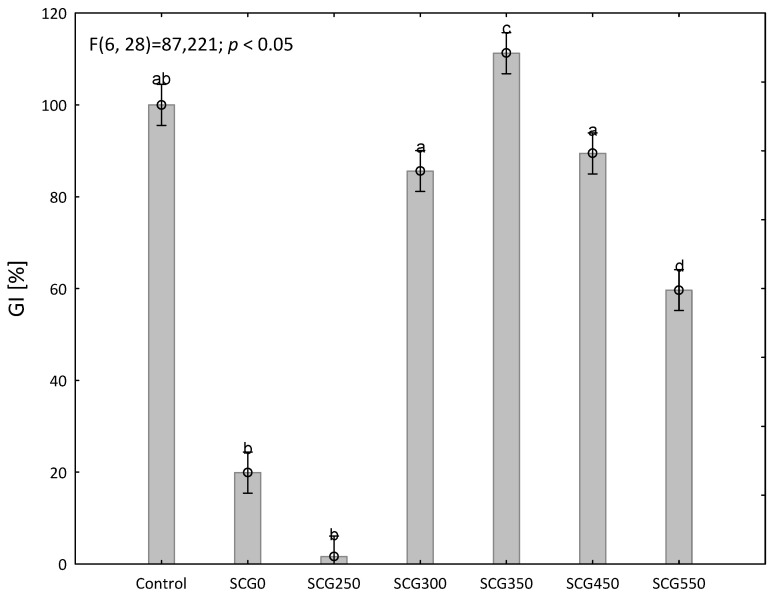
Phytotoxicity effect of SCG aqueous extracts on the germination of *Lepidium sativum* L. seeds after 48 h. Data are expressed as means of five independent bioassays (five replicates for each concentration (aqueous extracts) per bioassay) ± SE. Different letters (a–d) indicate significant differences between treatment effects when compared to the control (ANOVA, Tukey test, *p* < 0.05).

**Table 1 materials-15-06722-t001:** List of samples Spent Coffee Ground (SCG).

Sample	Description	Final Temperature
SCG0	Original dried SCG	-
SCG250	Torrefied SCG	250 °C
SCG300	Torrefied SCG	300 °C
SCG350	Torrefied SCG	350 °C
SCG450	Torrefied SCG	450 °C
SCG550	Torrefied SCG	550 °C

**Table 2 materials-15-06722-t002:** Coffee proximate and ultimate analysis.

Sample	Moisture (% wt.)	Carbo (% wt.)	Hydrogen (% wt.)	Oxygen (% wt.)	Nitrogen (% wt.)	Ash (% wt.)	Net Calorific Value (MJ·kg^−1^)
SCG0	8.22	50.26	6.29	31.37	2.21	1.59	19.74
SCG250	0.86	59.51	6.85	28.01	2.68	1.98	23.85
SCG300	1.46	71.04	6.73	14.34	3.39	2.95	29.09
SCG350	1.65	76.67	6.03	7.59	3.92	4.08	31.26
SCG450	2.64	77.94	3.55	5.08	4.41	6.35	29.88
SCG550	6.17	76.97	2.60	3.17	4.09	6.95	27.49

## Data Availability

Publicly available datasets were analyzed in this study. This data can be found here: [https://docs.google.com/spreadsheets/d/1E0n3cN-e8M0PJpC3y4p4-AngpvbY4Gnm/edit?usp=sharing&ouid=104516914322795462683&rtpof=true&sd=true].

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
