# Peer review of "Use of Spent Coffee Ground as an Alternative Fuel and Possible Soil Amendment"

_materials, 2022, doi:10.3390/ma15196722_

Round 1
Reviewer 1 Report
The current manuscript was investigated the spent coffee ground as an alternative fuel and soil amendment. In overall, the content of the manuscript was more looks like an experiment report instead of a technical manuscript. The English need to be polished by English native speaker or undergo English professional service. The novelty of this research needs to be highlighted. A comparative properties results between the current spent coffee ground and others from the literature review need to be compared. The desired properties for the fuel and soil amendment need to be provided. The research results for the fuel and soil amendment before and after added spent coffee ground biochar need to be presented as well. Authors need to explain the possible reasons on the sample SCG250 that resulted in an increment of phytotoxicity. Other comments include:
Inconsistent text – some text being bold throughout the manuscript especially the text in Stochiometric analysis methodology
There were two repeated subsections titled as Phytotoxicity test in section 2. Materials and Methods. In this section 2, a proper break down sub-section should be provided and the format should be tally with section 3.
Incorrect and inconsistent in-text citation i.e. [36] in line 214. It should be Fermoso and Mašek [36]
Authors should use Figure instead of Graph
Inconsistent term or typo error i.e. Graf
Graph 5: need to be improved in terms of the presentation and curve labelling in lagend
Graph 6 & 7 & 8: Provide proper axis label. There were 2 figures being named as Graph 7
Author Response
Hello,
thank you very much for your review.
I updated the document by all your points.
I let the manuscript to be checked and corrected by native speaker.
I highlighted the novelty of this research.
I checked the results against the accessible from literature review.
I estimated the desired properties (net calorific value)
I compared SCG as a fuel with usually used fossil fuels and as a soil amendment with control sample (without any SCG)
I discussed the reasoning for phytotoxicity increase for SCG250 - caused by release of toxic substances at 250°C - tanins, sulphur and caffeine
I checked and repaired all inconsistencies in the text.
I plan to deep dive more in these questioned areas to have more precise explanations of SCG behaviour when torrefied and publish it in next articles.
Best Regards and thank you again for your comments.
Lukas
Reviewer 2 Report
Please, unify the writing of units with respect to spacing etc.
With the increase in the carbon content at 550°C, why there is a decrease in the calorific value of the sample?
Graph 1 is not clear, I suggest changing the design of the graph.
The proximate and ultimate analysis of the sample can be tabulated.
The manuscript must be thoroughly checked for grammatical errors. Many sentences don’t have meaning. (E.g: L273 and many)
Why there is a sudden decrease in the CO2max at 300 to 350°C and an immediate increase at 450°C has to be explained clearly.
There are many typo errors in the manuscript that has to be seriously corrected. (E.g: L247, L271, L248)
The citations are not formatted uniformly.
Author Response
Dear colleague,
thank you very much for your review.
I checked all your comments and updated the document accordingly.
I unified whole document and let it be checked by a native speaker.
I updated the explanations of the figures.
With the carbon content increase, net calorific value is decreased for SCG 450 and SCG550 caused by decrease of hydrogen and oxygen share.
I updated slightly the Graph 1 to be better readable.
I added the table with average numbers of proximate and ultimate analysis
CO2max decrease for SCG300 and SCG350 was caused by decrease of hydrogen and oxygen. Meanwhile the increase of carbon led the increase of CO2max for higher temperatures.
I checked and updted the citations with citation software.
Best regards and thank you again for your valid comments.
Lukas
Reviewer 3 Report
Page 1, Line 21, 23 (and throughout the manuscript): Please insert a spacing between number and alphabet. Example: 60 kg, 9 g …
Page 1, Line 44: 20,9 MJ/kg or 20.9 MJ/kg?
Page 3, Line 106: “C. arabica” should be italicize.
Page 3, Line 109: The authors should state the duration of the process for drying SCG naturally under the sun. And until what moisture content? Below 5%?
Page 5, Line 172: Preferably to start a sentence with alphabet. “10…” should be replaced with “Ten…”
Page 5, Line 175: Preferably to start a sentence with alphabet. “2…” should be replaced with “Two…”
Page 5, Line 180: The authors are suggested to state “Lepidium sativum L.” in the methodology section, as stated on Page 11 Line 314.
Page 6, Line 212-216: These sentences require restructuring.
Page 7, Line 226: Sentence restructure. “…. influence directly ‘by’ the level of …”.
Page 10, Line 297: The authors mentioned that the total polyphenols in SCG is around 17.30% wt. Please state the approximate polyphenols in the torrefied SCG at 250°C, 300°C, 350°C, 450°C, and 550°C. These samples treated with various temperature will produce polyphenols in different amount. Thus, quantitative data of polyphenols is required to clarify the applicability of the SCG as soil amendment.
Page 11 Line 310: Please state the example of undesirable substances and elaborate how these undesirable substance effects the toxicity of the soil.
Page 11 Line 314: “Lepidium sativum L.” should be italicize.
Page 11 Line 325: Recheck. SCG550 or SCG350? Page 8 Line 248 stated as SCG 350.
References section:
It is preferable to cite the recent published journals, within 10 years (2012-2022). Some of the references is beyond 2012 (Number 10, 67 and 68). It is highly suggested that the authors replace and include below references in the lists, which is related to the research scope:
(1) Fermentation of Palm Oil Mill Effluent in the Presence of Lysinibacillus sp. LC 556247 to Produce Alternative Biomass Fuel. https://doi.org/10.3390/su132111915
(2) Renewable biomass feedstocks for production of sustainable biodegradable polymer. https://doi.org/10.1016/j.cogsc.2020.100412
Author Response
Dear colleague,
thank you very much for your revision,
I have updated the document by all your comments.
I have checked the document and corrected all typos, unified all values and units. I let native speaker to check the document.
I have better explained the material collection process.
I realize that polyphenol analysis is important. But unfortunately their determination was not the focus of this article, will do such analysis in next research.
Toxic substances that caused the toxicity increase for SCG250 are residues of caffeine, tannins and other substances.
I citted both proposed articles.
Thank you again for your valid comments and best regards,
Lukas
Reviewer 4 Report
In its current form, the manuscript entitled "Use of spent coffee ground as an alternative fuel and soil amendment" is more like a laboratory report rather than a scientific article. In my opinion, this work requires a more in-depth analysis of the researched materials. As it stands, Im not recommend the manuscript for publication
Author Response
Dear colleague,
thank you for your comments.
I tried as much as possible to update the document to make it more a scientific article instead of a laboratory report. Please check the results.
As this is my first publication posted to this journal I'd appreciate a more concrete comments how to improve, either this or any future articles.
Thank you again and Best Regards,
Lukas
Reviewer 5 Report
The manuscript submitted by Jeníček et al. describes the Use of spent coffee ground as an alternative fuel and soil amendment. The manuscript is presented well, the introduction and the results are written in good way. However,authors should carefully revise the manuscript based on the following comments before it can be considered further for publication.
1- In the introduction session, the novelty and significance of the work should be emphasised. In addition, the potential impact of the research and why it is important, compared to other research in this field or previous studies, should be discussed.
2- There are several abbreviations and it would be better to make a list rather than to run through the text.
3- Please pay attention to some typos.
4- Did the authors evaluate the effect of spent coffee ground concentration?
5- why there is no any characterization for Coffee, it is very important to confirm your material before using?
Author Response
Dear colleagues,
thank you for your comments.
I updated the document accordingly and let it be checked with a native speaker.
I emphasized the novelty and importance of this article.
I checked the results with other articles.
I unified and simplified the abbreviations
Unfortunately we did not evaluate the effect of different concentrations when checking the SCG as soil amendment. But would like to do in next reasearch.
I updated the characterization of coffee.
Thank you again for your valid comments, which helped me to update the document.
Best Regards,
Lukas
Round 2
Reviewer 1 Report
The authors had addressed the comments
Reviewer 2 Report
The manuscript was improved considerebly. It can now be accepted.
Reviewer 4 Report
The manuscript have some changes but in my opinion still its need high corrections to change for scientific paper.